# Current Advances in Mesenchymal Stem Cell Therapies Applied to Wounds and Skin, Eye, and Neuromuscular Diseases in Companion Animals

**DOI:** 10.3390/ani14091363

**Published:** 2024-04-30

**Authors:** Rosa Ana Picazo, Concepción Rojo, Jesus Rodriguez-Quiros, Alfredo González-Gil

**Affiliations:** 1Department of Physiology, School of Veterinary Medicine, Complutense University of Madrid, 28040 Madrid, Spain; rapicazo@ucm.es; 2Department of Anatomy and Embryology, School of Veterinary Medicine, University Complutense of Madrid, 28040 Madrid, Spain; rojosalv@vet.ucm.es; 3Department of Animal Medicine and Surgery, School of Veterinary Medicine, Complutense University of Madrid, 28040 Madrid, Spain; jrquiros@ucm.es

**Keywords:** mesenchymal stem cells, cell therapy, spinal cord injury, wound, skin disease, eye disease, dogs, cats

## Abstract

**Simple Summary:**

The search for alternative treatments is necessary for diseases where conventional therapies are ineffective. In recent years, therapies using mesenchymal stem cells have emerged as one of the most appropriate alternatives in regenerative medicine. Therapy with these types of cells is progressively increasing as a therapeutic option in veterinary medicine, leading to significant advances in treating certain pathologies. This review summarizes the current knowledge on mesenchymal stem cell therapies and their potential therapeutic and clinical effects on wound and skin, ocular, and neuromuscular diseases in dogs and cats.

**Abstract:**

Mesenchymal stem cells (MSCs) are considered a very promising alternative tool in cell therapies and regenerative medicine due to their ease of obtaining from various tissues and their ability to differentiate into different cell types. This manuscript provides a review of current knowledge on the use of MSC-based therapies as an alternative for certain common pathologies in dogs and cats where conventional treatments are ineffective. The aim of this review is to assist clinical veterinarians in making decisions about the suitability of each protocol from a clinical perspective, rather than focusing solely on research. MSC-based therapies have shown promising results in certain pathologies, such as spinal cord injuries, wounds, and skin and eye diseases. However, the effectiveness of these cell therapies can be influenced by a wide array of factors, leading to varying outcomes. Future research will focus on designing protocols and methodologies that allow more precise and effective MSC treatments for each case.

## 1. Introduction

In the last few decades, stem cell-based therapies have emerged as a highly promising area of research. Greater knowledge about the specific properties and mechanisms of action associated with these cells has resulted in significant progress on their clinical use as alternative treatments for a broad range of pathologies under criteria of safety and efficacy. High proliferation and self-renewal activity, along with the capacity for differentiation, are properties that make these cells optimal candidates for use in regenerative medicine and cell therapy [1]. Despite these promising advances, several concerns have limited their use, including the development of tumors in animal models, rejections of certain transplants, and ethical and social issues associated with the use of these cells, particularly those of embryonic origin [2].

Legislative and bioethical considerations have driven the development of cell therapy primarily in veterinary medicine rather than human medicine. Thus far, the utilization of these cells as an alternative therapy in veterinary medicine has yielded heterogeneous results. This variability can be attributed to a multitude of factors that impact treatment effectiveness, including the cell source, type and degree of injury, cell dosage, administration route, and patient characteristics. Further replication of various protocols and the inclusion of a larger patient cohort are essential to establishing tailored and effective protocols for different types of pathologies.

Mesenchymal stem cells (MSCs) are easily isolated from various tissues and possess the ability to differentiate into cells from diverse tissues [3]. This potential has positioned MSC-based therapy as a highly promising treatment for diseases where conventional treatment approaches fall short, including wounds and skin, eye, and neuromuscular diseases. This manuscript aims to provide veterinarians with a clinically-oriented review of the current knowledge on the use of MSC-based therapy as an alternative treatment for these conditions, aiding in decision-making regarding protocol suitability for dogs and cats.

## 2. Stem Cells and Classification

Stem cells can be defined as non-specialized cells with the ability for self-renewal and differentiation into multiple cell types, derived from a single cell (clonal property) upon receiving the appropriate triggers [2,4]. Stem cells are classified based on their differentiation potential and origin.

According to their potency or differentiation phase, stem cells can be divided as follows: (a) totipotent, cells which appear in the first stages of embryonic development and have the ability to divide and differentiate into cells of the entire organism, including both embryonic and extraembryonic structures; (b) pluripotent, which can differentiate into cells of the three germ layers (endoderm, ectoderm and mesoderm) and have the capacity to produce all cell types for all tissues and organs; (c) multipotent, which can differentiate into specific cell lineages and include MSCs; (d) oligopotent, which can differentiate into two or more cells types that belong to a specific class of tissue; (e) unipotent, with the capacity to differentiate into a single specific cell type, although they can undergo repeated division [1,2].

According to their origin, stem cells can be classified into the following: (a) embryonic stem cells, which are totipotent and/or pluripotent cells obtained from embryonic tissue; (b) fetal stem cells, which come from fetal tissue before birth; (c) adult or somatic cells, which are cells originating from the individual after birth; and (d) induced pluripotent stem cells (iPSCs), which are cells in which, after differentiation, pluripotency has been induced again through cellular reprogramming [5].

Adult stem cells exhibit less potency compared to the other three types of cells, as they can only differentiate into a limited number of cell types. However, adult stem cells are the most used in clinical cell therapy due to several advantages over embryonic and fetal stem cells, as they do not have ethical and legal conflicts, and with respect to iPSCs, since they do not present the tumorigenic potential associated with certain iPSC reprogramming used in animal models [6]. Adult stem cells exhibit other advantages, such as the fact that they can be obtained more easily since they are present in practically all body tissues. Furthermore, this allows them to be used autologously, thus avoiding immunological rejection problems [5]. Therefore, adult stem cells represent a highly promising cell source in current regenerative medicine.

## 3. Properties and Main Sources of MSCs

MSCs are immature and multipotent cells with a spindle-shaped morphology, like fibroblasts. MSCs offer several advantages over other stem cells, such as easy isolation from multiple tissues, a lack of ethical restrictions, and the ability to generate a large number of cells for clinical applications due to their high proliferative and self-renewal capacity, along with their capacity to differentiate into specific cell types of mesodermal origin [7,8]. Currently, MSCs are the most used in cell therapy for companion animals.

The Mesenchymal and Tissue Stem Cell Committee of the International Society for Cellular Therapy has established specific criteria to consider these cells as MSCs: first, that they are plastic-adherent under standard cell culture conditions; second, at least 95% of the cell population should express the markers CD73, CD90, and CD105 and, on the contrary, lack expression of hematopoietic surface markers such as CD11b or CD14, CD19 or CD79α, CD34, CD45, or the HLA class II markers; and third, under standard in vitro differentiation conditions, these cells should demonstrate the ability to differentiate into adipocytes, osteoblasts, and chondroblasts [9].

MSCs have the capability to stimulate cell division through the synthesis and secretion of trophic factors that act in a paracrine fashion, such as brain-derived neurotrophic factor (BDNF), vascular endothelial growth factor (VEGF), or insulin-like growth factor-1 (IGF-1) [10]. Moreover, MSCs exhibit immunomodulatory, anti-inflammatory, angiogenic, and anti-apoptotic properties that are crucial for tissue repair [11,12]. Therefore, MSCs are considered a highly promising alternative for tissue regeneration.

The anti-inflammatory and immunomodulatory properties of MSCs enable the improvement or suppression of the activity of various immune cells located in the injured area. This is achieved through the release of factors via cell-cell contact mechanisms. First, MSCs can directly interact with immune cells and local factors or release biomolecules such as cytokines, transforming growth factor beta (TGF-β), prostaglandin E2 (PGE2), or tumor necrosis factor alpha (TNF-α) that exert a paracrine action on cells from injured tissues [13]. Secondly, MSCs can secrete extracellular vesicles (EVs), such as exosomes, containing biomolecules that similarly modulate the action of immune cells [14], among other multiple signaling effects. The anti-inflammatory and immunosuppressive abilities of MSCs may be beneficial in certain pathologies, such as spinal cord injuries, where their administration could stimulate neuroprotection in the injured area [15,16].

Moreover, MSCs may also contribute to angiogenesis through the secretion of trophic factors such as VEGF and IGF-1 [12]. The secretion of anti-inflammatory molecules and trophic factors that stimulate tissue regeneration and angiogenesis facilitates improved delivery of oxygen and nutrients to injured tissue [17], which is essential for tissue repair.

Likewise, MSCs exhibit anti-apoptotic properties [12] that may contribute to the repair of injured areas, such as the spinal cord [18,19].

These therapeutic properties and the ability of MSCs to differentiate into specific cells of mesodermal origin, along with their effects derived from paracrine signaling [20], also allow multiple applications in other diseases. Thus, in wounds and certain skin diseases, MSC-based therapies might promote angiogenesis and neovascularization along with collagen synthesis, reepithelialization, and reduction of inflammation [21]. Additionally, the anti-inflammatory and immunomodulatory properties of these cells can be particularly useful in some eye diseases, such as keratoconjunctivitis sicca [22]. In this way, MSCs can promote both cell proliferation and viability at the injury site.

MSCs can be isolated from numerous tissues, and their properties can vary depending on the tissue source [23]. Currently, the two main sources of obtaining these cells have been bone marrow [24,25,26,27] and adipose tissue [25,28,29]. Other MSC sources are dental pulp [30,31], umbilical cord blood [25,31,32], Wharton’s jelly [25,33], amniotic fluid and placenta [34,35], synovium and synovial fluid [36], endometrium [37], and muscle [38]. Then, MSCs can differentiate in vitro into multiple cell types. This versatility allows for the application of MSCs in various pathologies, among which this review aims to highlight neuromuscular, skin, and ocular diseases (Figure 1).

Additionally, MSCs can be collected from the same individual (autologous MSCs) or from a donor of the same species (allogeneic MSCs). Autologous MSCs have some advantages, such as ease of obtaining and the lack of immunological rejection after infusion, but they may present some disadvantages, including cost and time of obtaining and issues related to the individual’s age or illness. Allogeneic MSCs can offer several advantages, such as allowing donor selection, multiple sources, and easy availability, although their main disadvantage may be the potential for immunological rejection. Currently, allogeneic MSC therapy is increasing in the clinical field, with these cells being considered clinically effective [39], although their clinical safety is still under debate, especially in repeated-dose treatments. The application of xenogeneic transplants using MSCs from a species other than that of the recipient is shown as another therapeutic alternative. These interspecies transplants may present some disadvantages, such as immune rejection problems, but could offer a practical alternative to expensive and inconvenient autotransplants [40].

## 4. Therapeutic Applications of MSCs

In recent years, MSCs derived from multiple tissues have been used in different cell therapies and regenerative medicine, yielding variable results. This review focuses on three groups of pathologies where the use of these cells is demonstrating more beneficial outcomes: neuromuscular diseases; wound and skin diseases; and eye diseases.

### 4.1. Neuromuscular Diseases

*Spinal cord injury* (SCI) is a common neurological disorder in companion animals, usually associated with an inflammatory process and complex pathophysiology, causing motor and sensory alterations that can lead to neurological loss and irreversible paralysis. The usual treatment in companion animals involves surgical stabilization or decompression, which includes the removal of diseased intervertebral disc material. However, recovery rates are variable and often unsatisfactory, failing to successfully restore neurological functions in the most severe cases [41]. Therefore, new therapeutic alternatives are required to reduce inflammation and promote tissue regeneration.

The characteristic anti-inflammatory and trophic activities of MSCs, along with their ability to stimulate axonal growth, remyelination, and cell proliferation, could be a stimulus for the regeneration process in damaged tissues [41,42]. However, to date, the results obtained in different studies display considerable variability due to several reasons. First, disease progressions between patients with experimentally induced lesions and those with natural or spontaneous lesions are often very different. Second, variability in results may also stem from different treatment approaches, such as the patient’s breed, age, sex, and weight; location and severity of injury; cellular sources; dose and route of cellular administration; allogeneic or autologous origin; use of MSCs included in scaffolds; or previous treatment. Minimizing this variability is essential for optimizing treatment efficacy. In dogs, SCI is commonly caused by trauma or by intervertebral disc herniation (IVDH).

Several clinical studies have demonstrated the therapeutic potential of MSCs in treating SCI in dogs. For instance, in two studies involving dogs with acute SCI [43,44], a single dose of 1 × 10^7^ allogeneic MSCs derived from adipose tissue (ad-MSCs) was administered into the lesion site following surgical decompression or hemilaminectomy. The administration of these cells led to faster locomotor recovery and reduced postoperative hospitalization, likely attributable to their anti-inflammatory effect and a possible improvement in the survival of endogenous nerve cells observed during the 6-month follow-up period [43].

On the other hand, several studies have evaluated the effect of these cells in dogs with chronic SCI. A pilot clinical study included six dogs with chronic SCI, where a single dose of 1 × 10^7^ allogeneic ad-MSCs labeled with ^99m^Tc was administered percutaneously to the injury site [45]. Results showed no adverse effects during the 16-week follow-up period, with three dogs showing improved locomotion, as assessed by the Olby scale, and one dog being able to walk without support.

Despite the progress observed in dogs with SCI treated with a single dose of MSCs, magnetic resonance imaging (MRI) did not show advancements in tissue regeneration in several clinical studies. In these investigations, dogs with chronic SCI were first subjected to decompressive surgery, and then a single dose of MSCs was inoculated at the site of injury [46,47,48,49,50]. Although significant improvements in movement and progressive recovery of some reflexes and intestinal and urinary bladder functions were observed, no evolution was indicated in the MRI images.

There is a case report involving a cat with a compression fracture where, after hemilaminectomy, a dose of 7 × 10^8^ autologous MSCs derived from bone marrow (bm-MSCs) was administered at the injured area. Seven days later, significant clinical improvement was observed, with progressive recovery of panicular reflexes, responses to superficial and deep painful stimuli, and partial restoration of intestinal and urinary bladder function at 75 days [51].

To date, the only study demonstrating regeneration of injured tissue after the application of a single dose of MSCs has been carried out in a dog with SCI [35]. In this study, after a previous decompression, 5 × 10^6^ MSCs derived from amniotic fluid were administered directly into the lesion. Approximately 15–16 weeks later, a clear improvement in the hind limb movements was observed, with the dog able to walk independently, although not perfectly. MRI images from this study indicated spinal cord regeneration.

With the aim of achieving greater tissue recovery and regeneration, recent clinical studies have focused on establishing multidose transplants at different time intervals instead of a single infusion. The application of two doses of MSCs with an interval of 7 days was indicated in two studies in dogs with chronic SCI [52,53]. In one study, two doses of 1.2 × 10^6^ ad-MSCs were administered along with low-density electrical stimulation [52], while in another study, 2 × 10^6^ allogeneic MSCs derived from canine dental pulp (dp-MSCs) were combined with electroacupuncture [53]. Both studies reported slight neurological or motor improvements, but the results were not very conclusive.

Another study using thirteen paraplegic dogs with IVDH without clinical improvement for at least 42 days after SCI also transplanted two doses of 5.0 × 10^6^ autologous neurogenically induced bm-MSCs into the spinal cord, with an interval of 21 days [54]. Although no adverse effects were observed during the 4- to 8-month follow-up, the most significant clinical progress only occurred two months after the second MSC transplant, with improvements noted in the gait score in six dogs and in proprioception and nociception for only two animals.

In an interesting clinical study by Branco et al. (2019), two dogs, one with chronic SCI (animal 1) and the other with subacute spinal injury (animal 2), underwent treatment with three infusions of autologous ad-MSCs and physiotherapy sessions [55]. The first infusion of 7 × 10^6^ was administered 30 days after the surgical procedure, followed by two subsequent infusions with a 3-month interval, administered both locally and intra-arterially, through the dorsal metatarsal artery. Animal 1 showed partial recovery of walking movement, while animal 2 achieved complete recovery. In addition to the 3-dose regimen of MSCs inoculated with wide time intervals, this study introduces some interesting features, such as the use of dual infusions, both local and intra-arterial. This last route, in addition to avoiding passage through the lungs, enables MSCs to target the injury site via chemotaxis, making it an efficient delivery route.

Another clinical study involving four dogs with chronic SCI who underwent previous decompression surgery utilized a 3-dose regimen of MSCs [56]. The first dose comprised 2 × 10^6^ allogeneic MSCs derived from the amniotic membrane (am-MSCs) topically administered on the lesion, followed by two additional doses epidurally after 15 and 45 days. This treatment approach resulted in neurological improvement, enhanced proprioceptive capacity, and restoration of nociception in some animals.

In two studies conducted by Sharun et al., one involving six dogs with compression fractures [57] and another with one dog exhibiting paraplegia associated with IVDH [58], four doses of allogeneic bm-MSCs (1 × 10^6^ cells/mL) were percutaneously infused at the injury site with a 15-day interval. Significant improvements in locomotor status and sensory functions were observed in all dogs.

Other authors administered four doses of 1 × 10^6^ allogeneic bm-MSCs intravenously (IV) over a month, combined with physiotherapy sessions, to four dogs with chronic SCI [59]. Up to 6 months of follow-up, no adverse effects were reported, and all dogs demonstrated clinical improvement according to the Olby scales, with two animals showing improvement in bladder function. Furthermore, in a study involving a larger number of clinical cases (44 dogs) with different degrees of SCI, conventional medical management was combined with the administration of allogeneic bm-MSCs (1 × 10^6^) every 15 days up to 45 days. Animals treated with MSCs showed significant improvements in functional recovery, proprioception, gait, bladder tone, and defecation control [60].

To date, only one study in dogs with SCI included up to six administrations, with no adverse effects during the 6-month follow-up [41]. In this research, four dogs with SCI resulting from traffic accidents, with previous surgical decompression, were treated with a first dose of 5 × 10^6^ allogeneic ad-MSCs to the nerve roots at the site of the injury plus another IV dose of 4 × 10^6^ cells/kg of weight, administered 30 min after surgery. Additionally, four additional doses of ad-MSCs were epidurally transplanted every 2 weeks after surgery. All dogs showed significant neurological improvements with ambulatory ability and normal urinary control in three animals.

Therefore, the absence of adverse effects and the greater degree of recovery obtained could justify transplantation with multiple doses of MSCs in patients with SCI. However, evaluating tissue regeneration through histopathological assessment of spinal tissue is very difficult to obtain in client-owned animals.

To improve tissue regeneration, several experimental trials combined the use of MSCs with other cells or factors with anti-inflammatory, antioxidant, or regenerative activity and achieved encouraging results in the canine model of SCI. Platelet-rich plasma is used as a vehicle for MSCs, facilitating tissue regeneration through the release of growth factors and cytokines [61,62]. Furthermore, additional studies have investigated the inclusion of other factors along with MSCs, such as recombinant methionyl human granulocyte colony-stimulating factor [63], BDNF and heme oxygenase-1 [64], heat shock proteins [65], or cocultured with Schwann cells [66], obtaining improvements in motor function and nerve regeneration at the site of injury. Furthermore, some experimental studies in canine models of SCI included MSCs within polymeric structures or scaffolds that are implanted at the site of the injury, resulting in significant advances in the regeneration of damaged tissue. Thus far, the material of these scaffolds used together with MSCs is variable, i.e., poly lactic-glycolic acid [67], matrigel [25,68], collagen/sulfate of heparin [69], or linear collagen [70], showing significant improvements in hindlimb locomotor recovery, neural regeneration, good integration into host tissue, and reduced fibrosis.

Therefore, transplantation with MSCs seems feasible and safe, with promising potential for clinical and regenerative medical applications in companion animals with SCI. Future treatments will probably be aimed at the use of MSCs embedded in scaffolds and their combination with factors to enhance anti-inflammatory and regenerative capacity. Multiple doses of MSCs should support all of these effects, especially when these cells are administered locally or through routes allowing infusion at the site of injury, such as the intra-arterial route. Future studies that combine cell therapies and tissue engineering will be necessary in a larger number of patients, especially with spontaneous lesions whose development is usually very different from that of animals with experimentally induced lesions.

### 4.2. Wound and Skin Diseases

Wounds and skin diseases are routinely diagnosed in veterinary practice. Treatment is usually complex and ineffective. Hence, the anti-inflammatory and regenerative properties associated with MSCs offer promising potential for addressing these cases. Extensive skin wounds often pose significant challenges in treatment and healing, with many conventional approaches proving ineffective.

The process of *wound* healing involves a series of sequential mechanisms aimed at repairing injured tissue. Many cases, especially large wounds, represent a major problem due to healing difficulties. In recent years, both experimental and clinical studies have suggested MSC treatment could modulate the local inflammatory response and promote cell replacement through paracrine mechanisms, thereby improving impaired wound healing.

A first clinical study was carried out on twenty-four dogs with acute and chronic wounds caused by sports activities or domestic injuries. Animals were treated with an initial dose of 3 × 10^7^ allogeneic ad-MSCs injected intradermally around wounds with an area of up to 10 cm^2^ (2 doses if larger), followed by a second dose on day 10 after the initial treatment. The results showed improved healing, with re-epithelialization observed in both types of wounds in 97% of treated dogs at 90 days [71].

In a second study conducted by Encino et al. (2020), a healthy dog with multiple bite wounds on the skin used a single dose of ad-MSCs along with antibiotic and anti-inflammatory treatment administered for 8 days. On the third day, 1 × 10^7^ allogeneic ad-MSCs were injected intradermally into some wounds. The findings revealed a greater regenerative capacity and an earlier and faster closure of the wounds treated with ad-MSCs compared to untreated control wounds. Additionally, treated wounds exhibited an absence of inflammatory infiltrates and the presence of multiple hair follicles [72].

In one study involving two dogs with large skin lesions that did not respond to standard treatments, MSCs isolated from human umbilical cord Wharton’s jelly were applied locally along with a poly (vinyl alcohol) hydrogel membrane to promote wound healing [73]. The wounds were infiltrated with 1 × 10^5^ cells/cm^2^ of lesion area. This treatment led to significant progress in skin regeneration, as evidenced by the reduction in ulcerated areas confirmed by histological analysis. In a more recent study by Humenik et al. (2023), seven dogs with complicated wounds of different etiologies that did not heal with conventional treatment were treated with a combination of antibiotics, non-steroidal anti-inflammatory drugs, and a local application of canine am-MSCs on the wound surface. The results showed a 98.47% reduction in wound surface area compared to 57.13% in the control group treated with conventional therapy [74].

Tissue bioengineering focused on regenerative medicine with the production of decellularized and/or recellularized skin scaffolds with MSCs could also constitute an innovative and valuable approach for treating complex wounds in companion animals [75]. A clinical study conducted on a dog with large bite wounds exemplifies this concept [76]. In this study, on a dog with two large bite wounds of approximately 70 cm^2^ each, a bioactive 3D matrix was employed to stimulate tissue regeneration around the wounds. One wound was treated with a matrix based on Tissucol fibrin glue, which contains a combination of adenoviral constructs with VEGF165 and fibroblast growth factor-2 (FGF-2) genes plus 3 × 10^6^ ad-MSCs. The other wound received the same composition but without MSCs. An improvement in tissue regeneration and wound recovery was observed in both cases, especially after MSC-based treatments.

*Canine atopic dermatitis* (AD) is a common immune-mediated skin disease with a high prevalence in companion animals. Current treatments exhibit great variability in their effectiveness and safety; hence, the immunomodulatory properties associated with MSCs make them a promising treatment alternative.

In a study conducted in twenty-two dogs with refractory AD who did not respond to conventional therapy, allogeneic ad-MSCs were IV administered at a dose of 1.5 × 10^6^/kg body weight [77]. The Canine Atopic Dermatitis Extent and Severity Index, version 4 (CADESI-04) scoring system, previously validated [78], showed that scores significantly improved one month after treatment and remained stable for at least 6 months of follow-up without adverse events. Additionally, a significant decrease in pruritus was observed one week after treatment.

More recently, a double-blind, placebo-controlled study [79] examined the efficacy of allogeneic ad-MSCs treatment in 15 dogs with AD randomly assigned to one of three groups: placebo, low-dose ad-MSCs (5 × 10^5^ cells/kg), and high-dose ad-MSCs (5 × 10^6^ cells/kg), injected subcutaneously at five body sites on days 0, 30, and 60. High-dose treatment resulted in significantly lower levels of miR-483, a potential prognostic marker for canine AD, compared with the placebo group at 90 days. The CADESI-4 scores showed decreasing trends in the high-dose group, indicating a decreased degree of pruritus and associated signs up to 30 days after the last administration [79].

In another study, sixteen dogs with AD were divided into three groups (i.e., mild, moderate, and severe) according to the CADESI-4 system. All animals received IV injections of 2 × 10^6^/kg body weight of allogeneic ad-MSCs on days 10, 31, and 52. The results showed significant clinical improvement, with a reduction in epidermal thickness observed in the moderate and severe groups throughout the 82-day follow-up without side effects [80].

A novel pilot study pioneered the use of multiple intramuscular (IM) doses of cryopreserved allogeneic ad-MSCs to treat twelve dogs diagnosed with AD. Before administration, these cells showed 90% viability after thawing. A dose of 0.5 × 10^6^ cells/kg ad-MSCs was IM-injected weekly for 6 weeks. Most disease symptoms were eliminated and/or decreased by 6 weeks, and both the Canine Atopic Dermatitis Lesion Index (CADLI) scores and the Pruritus Index showed significant improvements [81].

To our knowledge, only one clinical pilot study did not indicate significant improvements after MSC treatment in AD patients [82]. In this study, 1.3 × 10^6^ ad-MSCs/kg were IV administered to five dogs with AD, but the owner did not report a positive evolution of clinical signs or reduction in pruritus. However, there are some weaknesses in this study, such as the lack of characterization of the MSCs used.

In conclusion, preliminary results associated with the treatment of skin wounds and AD seem promising. However, future studies will be necessary to further investigate these treatments. Key considerations for future research include implementing double-blind analysis, increasing the number of animals studied, incorporating placebo control groups, and applying different protocols with different dosing regimens and intervals. Additionally, the incorporation of scaffolds may enhance the efficacy of these treatments when appropriate. However, the need for adequate funding is a critical factor that currently hinders the development of this research, making it complicated and slow.

Finally, this review includes a case report of a dog diagnosed with *pemphigus foliaceus*, an autoimmune skin disease characterized by acantholysis and refractory to steroids. Treatment was performed with 21 doses of ad-MSCs overexpressing cytotoxic T lymphocyte antigen 4 and/or naïve ad-MSCs over a period of 20 months with intervals of 2 to 8 weeks. The treatment showed a remission of the skin lesions, which made it possible to establish a regimen with low doses of prednisolone for 12 months [83].

### 4.3. Eye Diseases

Most of the MSC-based therapies in the canine ophthalmic clinic have been applied for two conditions: keratoconjunctivitis sicca (KCS) [22,84,85,86,87,88,89] and corneal ulcer (CU) [90,91,92], including descemetocele [93].

*KCS*, commonly referred to as “dry eye syndrome”, is a common eye disease in dogs characterized by inflammation of the lacrimal gland and decreased tear production. A quantitative or qualitative deficit of tears may lead to potential harm to the eye surface. Lifelong treatments for KCS consist of the application of immunosuppressive drugs such as cyclosporine A and tacrolimus [94]. There is ongoing research to explore alternative therapies aimed at reducing the lifelong dependence on drugs that suppress immune responses and inflammatory processes for treating patients with KCS [84]. The immunoregulatory properties of MSCs could be applicable to this pathology. Thus, a decrease in the concentrations of inflammatory markers was reported after MSC treatment in the conjunctiva and lacrimal gland, mainly CD4, IL-1, IL-6, and TNF-α [85]. Additionally, MSCs could aid in the regeneration of ocular tissues by secreting various bioactive trophic factors, which prompt adjacent parenchymal cells to initiate repair processes [85].

In the first uncontrolled study of MSC injection for naturally occurring KCS, twelve dogs were involved [86]. The patients were injected with 8 × 10^6^ allogeneic ad-MSCs: 5 × 10^6^ delivered around the main lacrimal gland and 3 × 10^6^ around the accessory lacrimal gland. After a 9-month follow-up period, nine out of twelve dogs achieved normal tear production without the need for topical immunosuppressive medications, and no adverse events were reported. In another study conducted in fifteen dogs, low doses of allogeneic ad-MSCs (1 × 10^6^) were directly injected into the lacrimal glands [80]. The authors did not observe adverse effects during the 12-month follow-up period. The eyes of dogs with mild–moderate KCS reverted to those of healthy eyes, whereas severe cases showed improvement in tear production and other clinical signs. In a third clinical study, three dogs were injected with ad-MSCs around the lacrimal glands at a high dose, i.e., 10 × 10^6^ cells around the main lacrimal gland and 5 × 10^6^ cells around the accessory gland [87]. Clinical improvement, particularly in tear production, was observed at 3-month follow-up after MSC treatment, with no further noted improvements. The frequency of administration of topical cyclosporine 2% and artificial tears was progressively reduced by up to half.

Periocular injection often requires sedation or general anesthesia. Hence, alternative routes have been tested in recent years. The use of topical treatment with MSCs, which involves absorption through the conjunctiva, has been a great advance for the treatment of canine KCS. In a study by Sgrignoli et al. (2019), topical application of 1 × 10^6^ allogeneic ad-MSCs was performed in the eyes of twenty-two dogs with KCS. Six months later, patients showed significant improvements in tear production, ocular pathology, conjunctival goblet cell concentration, and inflammatory cytokine profile, with no adverse effects reported [85]. Another study evaluated the therapeutic effect of topical application of canine ad-MSCs in twenty-three canine KCS patients [22]. All animals received 2 × 10^6^ cells weekly for six consecutive weeks, and complete ophthalmic examinations were performed at baseline, third, sixth, and ninth weeks. The results revealed an increase in the quantity and quality of tears, ameliorating the clinical signs and improving the quality of life of the dogs. The authors recommend topical treatment with ad-MSCs, especially for patients who do not respond well to immunosuppressive therapy. In a novel clinical trial, contact lenses were used as a vehicle for delivering MSCs [88]. Twenty dogs with KCS were divided into two groups (*n* = 10 each): a control group that received topical cyclosporine A, artificial tears, and antibiotic eye drops three times a day for 4 weeks; and the group that received a single dose of 2 × 10^6^ allogeneic fetal limbus-derived MSCs cultured in contact lenses. The following various ophthalmic parameters were measured in all patients: the Schirmer test, tear break-up time, impression cytology, Rose Bengal staining, and tear osmolarity. Both treatments were safe and effective and contributed to clinical improvement at a similar level in KCS dogs.

Finally, a recent study used the systemic route for MSC administration in twenty-eight dogs with KCS, divided into two groups (*n* = 14 each): a control group treated with cyclosporine A and a group treated with canine ad-MSCs [89]. In this study, a single dose of ad-MSCs was administered IV over 30 min, with the dosage varying according to body weight, i.e., 1 × 10^6^ cells for ≤10 kg, 2 × 10^6^ for 10–20 kg, and 3 × 10^6^ for 20–30 kg. After a 180-day follow-up period, dogs treated with ad-MSCs and with an initial Schirmer tear test result of 11–14 mm/min required only artificial tears on day 180, while dogs with a Schirmer tear test initial <11 mm/min required treatment with cyclosporine A on day 45. The authors noted that MSC-based treatment was more effective in the early stages of KCS, when the disease is less advanced and potentially reversible.

*CU* is a prevalent condition observed in veterinary ophthalmology clinics and can result from various factors, such as trauma, infection, or inflammation. It is characterized by a loss of the epithelium and part of the stromal layer, which leads to a loss of transparency in the affected area of the cornea. Without appropriate treatment, CU can progress to ocular perforation, causing potentially irreversible damage and loss of vision. Although there are limited clinical studies on MSC-based therapies in companion animals with CU, the available studies have shown very promising results.

In a clinical case involving a dog with CU and associated difficulties in healing due to diabetes, MSC application was proposed as an alternative treatment [90]. Two doses of 3 × 10^6^ canine ad-MSCs were administered topically with an interval of 48 h. The results showed an improvement in less than 14 days related to several parameters: blepharospasm, conjunctival hyperemia, mucopurulent ocular discharge, photophobia, corneal opacity, chemosis, pigmentation, neovascularization, and pain.

A comprehensive study based on treatment with MSCs was carried out in twenty-six dogs with different degrees of CU, where conventional treatment was ineffective [91]. A total of 3 × 10^6^ allogeneic ad-MSCs were divided into 12 doses of 250,000 cells, i.e., an initial application by subconjunctival route followed by 11 instillations (one per hour). The treatment was accompanied by antibiotic eye drops and tears to avoid possible infections and promote the action of the applied cells. Results indicated that 84.6% of cases showed complete healing of the ulcer within 14 days, without the need for surgical intervention.

In a third study carried out in a complicated clinical case of canine CU, local infiltration of 1 × 10^6^ canine dp-MSCs in four different points of the conjunctival tissue was established [92]. A notable improvement was reported after 12 h, with a neovascularization process observed a few days later. Complete healing occurred 25 days after the application of the MSCs, resulting in minimal scarring and the complete recovery of normal vision.

*Corneal descemetocele* is a serious consequence of progressive CU, characterized by a protrusion of Descemet’s membrane on the corneal surface. To date, there is only one clinical study carried out in a female Shih Tzu, in which 2 × 10^6^ allogeneic MSCs derived from a canine ovary were injected bilaterally into the conjunctiva [93]. Additionally, 5 × 10^5^ cells were administered topically to each eye. After 75 days, the lesion had completely healed in the absence of corneal opacity.

Despite the limited number of clinical studies, MSC-based treatment appears very promising for patients with KCS and CU. Other eye diseases, such as chronic superficial keratitis (CSK), chronic total uveitis (CTU), or feline eosinophilic keratitis (FEK), have also been the subject of clinical studies with MSC therapies, although the results are not clear and the number of clinical studies is even lower.

*CSK* is characterized by chronic inflammation of the corneal epithelium and anterior stroma, leading to corneal opacification and visual impairment. In a recent study, eight dogs with CSK were divided into two groups (*n* = 4 each): a control group that received conventional topical treatment with prednisone; and a group treated with 1 × 10^6^ allogeneic ad-MSCs injected subconjunctivally in the perilimbal region at 0 and 30 days [95]. The 110-day follow-up showed that after treatment with MSCs, no side effects were found, with a slight improvement, although less than that of conventional treatment. Additionally, the need for sedation during administration and the higher cost may further limit its use.

There is also a single clinical study published that includes MSC-based treatment in dogs with *CTU*, a serious complication of ophthalmic surgery that may compromise ocular functionality and structure [96]. The study was carried out in four dogs diagnosed with spontaneous total uveitis, and doses between 1 and 1.5 × 10^6^ MSCs (the source of the cells was not indicated) were administered by different routes. The authors reported a better recovery process when the MSCs were administered by the following routes: Tenon’s capsule, intravitreal, or anterior chamber of the eye, with the subconjunctival route being ineffective. During the 14-day follow-up period, the interruption of the acute inflammatory process and the restoration of transparency in the anterior chamber of the eye were observed. Although MSC-based treatment recovers damaged structures and reduces the inflammatory process, it must be considered that the different routes of administration require certain clinical conditions and are more invasive and traumatic.

*FEK* is a chronic keratopathy caused by an immune-mediated response to an unknown antigenic stimulus. Conventional treatment consists of topical anti-inflammatory agents (corticosteroids or megestrol acetate) and/or immunosuppressants (cyclosporine A). A clinical study investigated the safety and therapeutic effects of subconjunctival implantation of feline allogeneic ad-MSCs around ocular surface lesions in five cats with refractory FEK [97]. Two doses of 2 × 10^6^ cells were applied two months apart. During the 11-month follow-up period, no systemic or local complications were detected. Furthermore, clinical signs showed resolution of the corneal and conjunctival lesions. Despite the limitations of this study, such as the lack of a control group and the small number of animals treated, MSC-based therapy in FEK appears effective.

The anti-inflammatory and regenerative properties of MSCs make treatment a suitable alternative with very promising results in different eye diseases. Even though there are currently no clinical studies related to MSC-based treatment for other eye diseases, such as canine glaucoma, it is evident that research will advance further in this direction [98]. Despite not being the subject of this review, it is worth mentioning that in recent years, the use of EVs derived from MSCs has been promoted with encouraging results in eye diseases where conventional treatments have been ineffective, such as retinal degeneration in dogs [99]. Future studies that include a larger number of animals and different doses and routes of administration will be necessary to establish appropriate protocols for each ocular lesion. As mentioned previously, this will not be possible without adequate funding to support the dissemination of MSC-based treatments.

## 5. Future Approaches to MSC-Based Therapy in Veterinary Medicine

The rapid increase in MSC-based therapies in domestic animals has prompted numerous studies utilizing treatments and protocols with varying levels of scientific rigor. This diversity makes it challenging to conduct realistic comparisons between different studies. Therefore, it is necessary to standardize key parameters related to clinical stem cell research in animals to improve the quality of studies and provide greater confidence in this therapeutic approach [100].

Future advances in regenerative medicine should prioritize developing therapeutic strategies tailored to promote healing in specific injured tissues. However, the high diversity and heterogeneity found in MSC-based therapies present challenges in defining and standardizing treatment protocols. On the one hand, MSCs present differences in terms of phenotype, multilineage potential, and proliferation activity depending on the tissue source [101,102], even with differences depending on the selected region within the same tissue [103]. Moreover, variability within donors, in terms such as health condition or age [104], or the particularities of the patient or the injured area might also influence the effectiveness of MSC therapy. Greater precision and refinement in isolation, characterization, and differentiation techniques will contribute to the knowledge of the most appropriate MSC phenotypes for each tissue or lesion [105]. Furthermore, the potency, persistence, and viability of MSCs may also be influenced by the route of administration. Studies in laboratory animals have demonstrated that administering MSCs by subcutaneous or intraperitoneal routes showed greater efficacy and persistence compared to other commonly used routes, such as intravenous administration [106]. Upcoming research should also consider the potential for a cell-free therapeutic alternative via the MSC secretome. Extracellular vesicles, microvesicles, and exosomes exert numerous biological actions, such as anti-inflammatory, anti-apoptotic, and immunomodulatory effects. Furthermore, the secretome is highly biocompatible, less immunogenic than the MSCs themselves, and can even be designed to allow us to act on target cells or tissues in a specific way [107]. Future treatment protocols need to consider these factors comprehensively and implement them in larger animal studies. Standardizing these protocols is essential to ensuring the effectiveness and reproducibility of MSC-based treatments across different clinical applications. Once again, the economic component is a critical factor hindering progress in this field. The lack of an adequate financial model for research and treatment with MSCs makes it difficult to conduct multicenter studies and their clinical applications, especially in therapies involving autologous cells. Consequently, allogeneic, and xenogeneic cell therapies emerge as more reliable alternatives to the autologous route since they are less expensive.

Finally, it will be necessary to establish a legislative framework that allows ethical control in the handling of stem cells as therapy in companion animals. The interaction between different international regulatory groups is necessary to establish consensus guidelines and standards that guarantee the development of safe and effective cell therapies within an international regulatory framework [108]. These guidelines should be followed by the clinical veterinarian as well as by the different studies that propose the therapeutic use of these cells.

## 6. Conclusions

There is no doubt about the immense potential of MSC-based therapies for regenerative medicine. However, multiple factors can influence its effectiveness, hence the great heterogeneity observed in the results after MSC administration. Despite the current lack of knowledge about treatment with these cells, the first positive results are already being obtained. The advances found after cell therapy in some diseases, such as spinal cord injuries, wounds, and skin and eye diseases, are probably just the beginning of the future and important milestones that will be achieved with the application of these treatments in multiple pathologies. We are confident that the future of research with MSC-based therapies will bring important benefits to diseases in which traditional therapies have numerous limitations and lack effectiveness. This will require a greater understanding of the biology and physiological mechanisms of these cells, a more precise control of their differentiation process, and a greater knowledge of the different factors and mechanisms that influence the effectiveness of these cell therapies. In this way, protocols and methodologies can be designed and updated that allow treatments to be more effective and precise for each disease.

## Figures and Tables

**Figure 1 animals-14-01363-f001:**
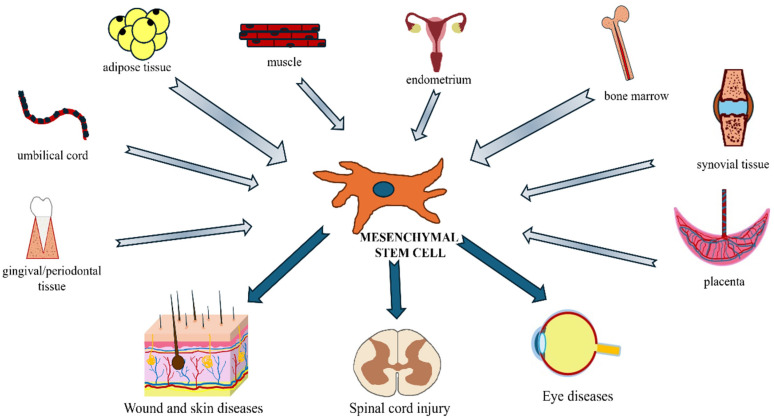
Main sources of mesenchymal stem cells and some clinical applications in companion animals.

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
