# Peer review of "Current Advances in Mesenchymal Stem Cell Therapies Applied to Wounds and Skin, Eye, and Neuromuscular Diseases in Companion Animals"

_animals, 2024, doi:10.3390/ani14091363_

Round 1

Reviewer 1 Report

Comments and Suggestions for Authors

The manuscript is well-written and describes the general characteristics of stem cells and their clinical applicability in some pet pathologies. However, the relevance that the secretome of these cells would have in their therapeutic potential was omitted. The manuscript would thus be enriched if this point is considered in its development.

line 179: use breed instead of race

Comments on the Quality of English Language

No comments

Author Response

Response to Reviewer 1 Comments

Thank you very much for taking the time to review this manuscript. We would like to thank the Reviewer for the positive and constructive evaluation of our work. We have considered all the comments in detail and have modified the manuscript where appropriate and/or provided a brief response below.

Point-by-point response to Comments and Suggestions for Authors

Comments 1: The manuscript is well-written and describes the general characteristics of stem cells and their clinical applicability in some pet pathologies. However, the relevance that the secretome of these cells would have in their therapeutic potential was omitted. The manuscript would thus be enriched if this point is considered in its development.

Response 1: We thank referee for this comment. The next paragraph has been included in the section “5. Future approach on MSCs-based therapy in veterinary medicine” corresponding to the new version of the manuscript: “Upcoming research should also consider the potential for a cell-free therapeutic alternative, via the MSC secretome. Extracellular vesicles, microvesicles and exosomes exert numerous biological actions, such as anti-inflammatory, anti-apoptotic and immunomodulatory effects. Furthermore, the secretome is highly biocompatible, less immunogenic than the MSCs themselves and can even be designed to allow us to act on target cells or tissues in a specific way [108]. (Page 12, lines 562-567)

Comments 2: line 179: use breed instead of race

Response 2: According to the reviewer “breed” has been included instead of “race” (Page 5, line 183)

We thank reviewer 1 for her/his valuable comments on the contents of our manuscript that have contributed to improve its quality.

Reviewer 2 Report

Comments and Suggestions for Authors

Thank you for this review on MSC use. Please see some grammatical and typographical issues outlined below:

Line 16: Number agreement “...this type of cells...” should be “...these types of cells...” 

Line 18: Number agreement “...stem cells therapies...” should be “...stem cell therapies...” 

Line 21: Grammar issue “...easy obtaining...” should be “...ease of obtaining them...” 

Line 56: “...veterinarian...” should be “...veterinarians...” 

Line 56: “...a clinically-oriented, a review...” should be “...a clinically-oriented review...” 

Line 143: Did you mean “versatility” instead of “versability”? 

Line 193: Please edit to remove “So,” at the beginning of this sentence. 

Line 193-194: Please provide the dose of MSCs used in this study, or clarify that dose was not indicated 

Line 302: “...addressing in these cases.” should be “...addressing these cases.” 

Line 316: I think you meant “single dose” not “simple dose” here 

Line 450: Typo “tewnty 

General comment: for the studies in which MSC sources were not indicated, did you attempt to contact the authors to provide this information?

Comments on the Quality of English Language

Please see the line by line comments to correct minor English language issues identified.

Author Response

Response to Reviewer 2 Comments

Thank you very much for taking the time to review this manuscript. We would like to thank the Reviewer for the positive and constructive evaluation of our work. We have considered all the comments in detail and have modified the manuscript where appropriate and/or provided a brief response below.

Thank you for this review on MSC use. Please see some grammatical and typographical issues outlined below:

Comments 1: Line 16: Number agreement “...this type of cells...” should be “...these types of cells...” 

Response 1: It has been properly changed in the text. (Page 1, lines 15-16).

Comments 2: Line 18: Number agreement “...stem cells therapies...” should be “...stem cell therapies...” 

Response 2: Number agreement was corrected (Page 1, line 18).

Comments 3: Line 21: Grammar issue “...easy obtaining...” should be “...ease of obtaining them...” 

Response 3: Grammar issue was corrected (Page 1, line 21).

Comments 4: Line 56: “...veterinarian...” should be “...veterinarians...” 

Response 4: Plural of veterinarian (veterinarians) was added (Page 2, line 57).

Comments 5: Line 56: “...a clinically-oriented, a review...” should be “...a clinically-oriented review...” 

Response 5: Corrected (Page 2, line 57).

Comments 6: Line 143: Did you mean “versatility” instead of “versability”? 

Response 6: Thank you for the comment. Yes, versatility is correct. (Page 3, line 143)

Comments 7: Line 193: Please edit to remove “So,” at the beginning of this sentence. 

Response 7: “So” was deleted from the text. (Page 5, line 197).

Comments 8: Line 193-194: Please provide the dose of MSCs used in this study, or clarify that dose was not indicated. 

Response 8: Thank you for this comment. The dose of MSCs used in the study was included in the new version of the manuscript: “… dose of 1 × 107 allogeneic ad-MSCs labelled with 99mTc was administered…” (Page 5, line 198).

Comments 9: Line 302: “...addressing in these cases.” should be “...addressing these cases.” 

Response 9: Corrected, “in” was deleted (Page 7, line 307).

Comments 10: Line 316: I think you meant “single dose” not “simple dose” here. 

Response 10: Yes, thank you for this comment; “single dose” was included instead of “simple dose” (Page 7, line 323).

Comments 11: Line 450: Typo “tewnty” 

Response 11: Corrected; “twenty” instead of “tewnty” (Page 10, line 452).

Comments 12: General comment: for the studies in which MSC sources were not indicated, did you attempt to contact the authors to provide this information?

Response 12: Thank you for the comment. No, we hadn't really consulted it. In accordance with the reviewer's comments, we have now consulted the authors of the two articles (references 83 and 93 in the initial manuscript; 87 and 97 in the revised manuscript) in which we had noted that the authors had not indicated the source of the MSCs.

In one of the articles [87], we had overlooked it, but they indicated the source (adipose tissue: ad-MSCs) that has now been included in the corrected version of the manuscript (Page 9, line 426). Sorry for the mistake. Regarding the other article [97], we have tried to contact the author (t_sav4uk@ukr.net) but so far, we have not received a response. They are researchers who belong to the National University of Life and Environmental Sciences of Ukraine, in Kyiv (Ukraine), so they may have problems due to the difficult situation in the country.

We thank referee 2, for her/his valuable comments that we have considered to improve the quality of this manuscript. We are keen to incorporate any other suggestion on it.

Reviewer 3 Report

Comments and Suggestions for Authors

Animals (ISSN 2076-2615)

animals-2982555

Current advances in mesenchymal stem cell therapies applied to wounds and skin, eye, and neuromuscular diseases in companion animals

Rosa Ana Picazo , Concepción Rojo , Jesus Rodriguez-Quirós , Alfredo González-Gil *

I congratulate he authors on a needed review on some of the current uses of stem cells in regenerative medicine in companion animals. Although only discussing skin, eye and neuromuscular uses of stem cells, readers will see the benefits of cell therapy.

I make the following recommendations based on my review of the manuscript and my experience and knowledge of the topic.

Line 21: “…their easy obtaining from various tissues…” This phrase needs to be rewritten for clarity.

Line 28: “Heterogeneous” may need to be replaced with a more appropriate word here.

Line 42: “…rejections in…” should be “…rejection of…”.

Line 56: Should be “…to provide veterinarians with a clinically-oriented review…”.

Line 84: “…do not have ethical and legal conflicts…” You may want to mention lack of tumorigenesis with adult stem cells.  

Line 85: “…they do not present the potential mutational effects that lead…”  Not all IPSCs promote tumor formation.

Line 86: If this is the initial use of iPSC, please define the abbreviation.

Line 116: “…paracrine molecules such as  cytokines,…”

Line 117-119: EVs have multiple signaling effects, not limited to immune modulation.

Line 122-123: “…the ability of MSCs to differentiate…” From my reading of your reference, MSCs' angiogenesis potential comes from their secretory properties rather than from their differentiation capacity. Vascular damage is another matter.

Line 130-137: Limiting the effects of MSCs on cell differentiation for wounds and skin conditions while limiting eye effects to EVs may limit MSCs' mechanism of activity too narrowly. Although these were citations, it is fairly well documented that paracrine signaling is a major factor in MSC effectiveness.

Line 140-143: Abbreviations of MSC sources are used in other areas of the paper and listed here. Although some abbreviations are defined in the text later in the paper, others are not. Please define.(Lines 311,316,319,334,348…)

Line 157: The clinical safety of allogeneic MSCs is still being debated, especially with repeated doses.

Line 161: See comment on line 28.

Line 186: The use of “inoculated” may be misunderstood by most readers. The choice of another word may be wise.

Line 198-200: Does this comment refer to the prior paragraph (citation 43) or to the following citations (44-48)?

Line 232-270: In these paragraphs, different routes of stem cell application are mentioned. Research in lab animals shows that intramuscular, subcutaneous, and intraperitoneal injections of MSCs may have advantages in some conditions. (See Mesenchymal stromal cell therapeutic potency is dependent upon viability, route of delivery, and immune match by

Jayeeta Giri and Jacques Galipeau (2020). This may need to be touched on either here or in the discussion.

Line 272-276: “However,…transplanted stem cells.” Please rewrite. Histopathological evaluation of spinal tissue is difficult (or impossible) to obtain in client-owned animals.

Line 276-282: “In this context…” I don’t see the context in which the following paragraph is referred.

Line 277-279: “…the use of MSCs…canine model of SCI.” Platelet-rich plasma is commonly used in conjunction with MSCs, and other research has documented the benefits.

Line 294-296: “Multiple subsequent…site of injury.” Very confusing sentence. “Intra-arterial transplants”?? Please clarify and rewrite.

Line 322-325: The entire paragraph may need to be moved to the initial paragraph of the section (Line 303). It seems out of place.

Line 327: “MSCs isolated from human umbilical cord…” You are introducing xenogeneic use here. Do you want to introduce that concept and potential to the novice reader?

Line 345-349: Very confusing sentence. Please rewrite for clarity.

Line 385-391: I completely agree with your recommendation for further investigation of MSC treatments of AD. One critical factor that you did not mention was the necessity of adequate funding of the research.

Line 402: Define KCS abbreviation.

Line427-428: “…with not complete…topical therapy.” Please rewrite for clarity.

Line 493: “Other eye diseases…” Please give examples. Are these case studies or case series?

Line 527-536: Same comment as to Line 385-391.

Line 532: “…microvesicles…” You may want to use extracellular vesicles (EVs) to be consistent with Line 117-118.

Line 537-564: Your discussion in this section is valid, but the reality of the situation is that most of the current application of MSCs is by individual veterinarians on client-owned animals. Case studies and case series studies are considered to be lower-quality research but are valuable research results, none-the-less. Multi-center, double-blind, placebo-controlled clinical trials cannot be done until financing is made available. Autologous use cell therapy does not have a viable financial model. Allogeneic and Xenogeneic use is becoming more common and could have a broader funding appeal to corporate sources of support.

Line 568: “…inoculation…” Same comment as Line 186.

Comments on the Quality of English Language

There are some areas that have been noted in the comments that are awkward and should be revised. There were a few word choices that may need to be changed.

Author Response

Response to Reviewer 3 Comments

Thank you very much for taking the time to review this manuscript. We would like to thank the Reviewer for the positive and constructive evaluation of our work. We have considered all the comments in detail and have modified the manuscript where appropriate and/or provided a brief response below.

I congratulate the authors on a needed review on some of the current uses of stem cells in regenerative medicine in companion animals. Although only discussing skin, eye and neuromuscular uses of stem cells, readers will see the benefits of cell therapy.

 I make the following recommendations based on my review of the manuscript and my experience and knowledge of the topic.

Comments 1: Line 21: “…their easy obtaining from various tissues…” This phrase needs to be rewritten for clarity.

Response 1: According to the referee, the phrase was rewritten for clarity: “…their ease of obtaining them from various tissues…” (Page 1, line 21).

Comments 2: Line 28: “Heterogeneous” may need to be replaced with a more appropriate word here.

Response 2: The term “heterogeneous” was replaced by “varying”. (Page 1, line 28).

Comments 3: Line 42: “…rejections in…” should be “…rejection of…”.

Response 3: Corrected (Page 1, line 42).

Comments 4: Line 56: Should be “…to provide veterinarians with a clinically-oriented review…”.

Response 4: The sentence was corrected in accordance with the referee´s comment (Page 2, line 57).

Comments 5: Line 84: “…do not have ethical and legal conflicts…” You may want to mention lack of tumorigenesis with adult stem cells.  

Response 5: Thank you for the comment. In this paragraph first we want to highlight the advantages of using adult stem cells compared to embryonic and fetal stem cells in which the main problems are related to ethical and legal conflicts. Then, we related the lack of tumorigenesis related with adult stem cells by comparing it with some studies where tumorigenicity was associated with iPSCs reprogramming: “...they do not present the tumorigenic potential associated with certain iPSCs reprogramming used in animal models [6]”. (Page 2, lines 86-87).

Comments 6: Line 85: “…they do not present the potential mutational effects that lead…”  Not all IPSCs promote tumor formation.

Response 6: We thank referee for this comment which we completely agree with. In fact, the use of iPSCs as cell therapy is becoming increasingly common and safe. The term "certain" has been included to indicate that not all iPSCs cause this problema (Page 2, line 86). If the reviewer considers it appropriate, a new sentence could be included indicating this assessment.

Comments 7: Line 86: If this is the initial use of iPSC, please define the abbreviation.

Response 7: The initial use of iPSC is located in the previous paragraph (Page 2, Line 79) where we have included the full name: "...d) induced pluripotent stem cells (iPSCs), which are cells...". Anyway, thanks for the comment.

Comments 8: Line 116: “…paracrine molecules such as cytokines,…”

Response 8: This sentence has been rewritten in the revised manuscript (Page 3, Lines 116-119).

Comments 9: Line 117-119: EVs have multiple signaling effects, not limited to immune modulation.

Response 9: We completely agree with the referee´s comment. Undoubtedly, the potential of the molecules included within these extracellular vesicles is enormous and they are emerging as one of the main alternative therapies for numerous diseases. We have included a brief clarification at the end of the sentence to indicate that they have numerous biological effects: “…, among other multiple signaling effects.” (Page 3, line 121). A paragraph explaining the general effects of the MSC secretome has been included in the new version of the manuscript (Page 12, lines 562-567). If the reviewer considers it insufficient, we could incorporate more text explaining other effects associated with these molecules.

Comments 10: Line 122-123: “…the ability of MSCs to differentiate…” From my reading of your reference, MSCs' angiogenesis potential comes from their secretory properties rather than from their differentiation capacity. Vascular damage is another matter.

Response 10: We agree with the comment, and to avoid confusion we have deleted from the text the sentence related to the differentiation capacity of MSCs towards vascular smooth muscle and endothelial cells. (Page 3, lines 124-125).

Comments 11: Line 130-137: Limiting the effects of MSCs on cell differentiation for wounds and skin conditions while limiting eye effects to EVs may limit MSCs' mechanism of activity too narrowly. Although these were citations, it is fairly well documented that paracrine signaling is a major factor in MSC effectiveness.

Response 11: Once again we agree with the referee's comment. The effects derived from paracrine signaling are essential in the therapeutic effect of MSCs. We have added in the sentence that along with the differentiation capacity, its paracrine activity also has a great influence on its therapeutic effects, supported with a new reference [20]: “…the ability of MSCs to differentiate into specific cells of mesodermal origin along with their effects derived from paracrine signaling [20]…” (Page 3, lines 130-131). The current review is fundamentally focused on the direct effects of MSCs without including those derived from their secretion of extracellular vesicles, which alone could lead to another review. However, due to the importance of the paracrine effects in these cells, a paragraph explaining the general effects of the MSC secretome has been included in the corrected version of the manuscript (Page 12, lines 562-567) as we have indicated above.

Comments 12: Line 140-143: Abbreviations of MSC sources are used in other areas of the paper and listed here. Although some abbreviations are defined in the text later in the paper, others are not. Please define (Lines 311,316,319,334,348…)

Response 12: Thank you for the comment. It was really a doubt that arose among the authors during the manuscript drafting process. Ultimately, we opted to introduce the abbreviation later, coinciding with the first mention of each tissue source from which the MSCs were derived. In this paragraph we finally decided that we only list the main sources. For example: MSCs derived from adipose tissue (ad-MSCs) (Page 5, line 190); bone marrow (bm-MSCs) (Page 5, line 210); dental pulp (Page 5, line 226-227); and amniotic membrane (am-MSCs) (Page 6, line 249). This last abbreviation (am-MSCs) was included in the revised manuscript. If the referee considers it appropriate, we would be willing to include the abbreviations in the initial paragraph (lines 139-143), with subsequent usage limited to the abbreviations throughout the rest of the text.

Comments 13: Line 157: The clinical safety of allogeneic MSCs is still being debated, especially with repeated doses.

Response 13: Thank you for the comment.  While many studies suggest the safety and efficacy of therapeutic use with these cells, further research is still required to substantiate this safety. According to the referee's comment, we have rewritten this last sentence adding "these cells being considered clinically effective [39], although their clinical safety is still under debate, especially in repeated dose treatments." (Page 4, lines 157-159).

Comments 14: Line 161: See comment on line 28.

Response 14: “heterogeneous” was replaced by “variable”. (Page 4, line 165).

Comments 15: Line 186: The use of “inoculated” may be misunderstood by most readers. The choice of another word may be wise.

Response 15: The term “inoculated” was replaced by “administered”. (Page 5, line 190).

Comments 16: Line 198-200: Does this comment refer to the prior paragraph (citation 43) or to the following citations (44-48)?

Response 16: Although it is a general comment regarding the majority of studies carried out in dogs with SCI, we relate it to the following paragraph where we include the references (44-48 in the initial manuscript; 46-50 in the revised manuscript) that expressly indicate that MRI showed no signs of tissue regeneration. (Pag 5, lines 202-208).

Comments 17: Line 232-270: In these paragraphs, different routes of stem cell application are mentioned. Research in lab animals shows that intramuscular, subcutaneous, and intraperitoneal injections of MSCs may have advantages in some conditions. (See Mesenchymal stromal cell therapeutic potency is dependent upon viability, route of delivery, and immune match by Jayeeta Giri and Jacques Galipeau (2020). This may need to be touched on either here or in the discussion.

Response 17: We appreciate the reviewer's suggestion, which we believe adds clarity and improves the manuscript. The following paragraph (and the recommended reference) has been included in section 5. Future approach on MSCs-based therapy in veterinary medicine: "Furthermore, the potency, persistence, and viability of MSCs may also be influenced by the route of administration. Studies in laboratory animals have demonstrated that administering MSCs by subcutaneous or intraperitoneal routes showed greater efficacy and persistence compared to other commonly used routes, such as intravenous administration [107]”. (Page 12, lines 558-562).

Comments 18: Line 272-276: “However,…transplanted stem cells.” Please rewrite. Histopathological evaluation of spinal tissue is difficult (or impossible) to obtain in client-owned animals.

Response 18: In response to the referee's comments, the paragraph has been revised as follows: “However, evaluating tissue regeneration through histopathological asessment of spinal tissue is very difficult to obtain in client-owned animals”. (Page 6, lines 276-278).

Comments 19: Line 276-282: “In this context…” I don’t see the context in which the following paragraph is referred.

Response 19: Thank you for the comment.  We aimed to link the previous comment on tissue regeneration evaluation with the subsequent paragraph in which we commented on the combination of MSCs with other factors to improve tissue regeneration. However, we agree with the referee that it may not be understood. Therefore, we have rewritten this paragraph as follows: “To improve tissue regeneration, several experimental trials combined…”. (Page 6, line 279).

Comments 20: Line 277-279: “…the use of MSCs…canine model of SCI.” Platelet-rich plasma is commonly used in conjunction with MSCs, and other research has documented the benefits.

Response 20: We agree with the referee's comment regarding the significance of using platelet-rich plasma as a vehicle for MSCs to augment tissue regeneration. We have included the following sentence in the new version of the manuscript:” Platelet-rich plasma is used as a vehicle for MSCs facilitating tissue regeneration through the release of growth factors and cytokines [61,62]. Furthermore, additional studies have investigated the inclusion of other factors along with MSCs...”. (Page 6, lines 281-284).

Comments 21: Line 294-296: “Multiple subsequent…site of injury.” Very confusing sentence. “Intra-arterial transplants”?? Please clarify and rewrite.

Response 21: We appreciate the feedback and have revised the sentence for clarity: “Multiple doses of MSCs should support all of these effects, especially when these cells are administered locally or through routes allowing infusion at the site of injury, such as the intra-arterial route.”. (Page 7, Lines 298-300).

Comments 22: Line 322-325: The entire paragraph may need to be moved to the initial paragraph of the section (Line 303). It seems out of place.

Response 22: We have included this paragraph to provide context for the treatment of large injuries, distinguishing them from smaller wounds discussed earlier. In agreement with the referee and for better understanding, we have moved the sentence “Extensive skin wounds often pose significant challenges in treatment and healing, with many conventional approaches proving ineffective” to the initial paragraph of the section (Page 7, lines 307-309). To avoid redundancy, we removed the sentence "Some studies have explored the use of MSCs due to their regenerative properties, which could directly benefit wound healing" from the text.

Comments 23: Line 327: “MSCs isolated from human umbilical cord…” You are introducing xenogeneic use here. Do you want to introduce that concept and potential to the novice reader?

Response 23: We agree with referee comment. The following paragraph has been added: “The application of xenogeneic transplants using MSCs from a species other than that of the recipient is shown as another therapeutic alternative. These interspecies transplants may present some disadvantages, such as immune rejection problems, but could offer a practical alternative to expensive and inconvenient autotransplants [40].” (Page 4, lines 159-162).

Comments 24: Line 345-349: Very confusing sentence. Please rewrite for clarity.

Response 24: The paragraph was rewritten, and the next sentence was included: “An improvement in tissue regeneration and wound recovery was observed in both cases, especially after MSCs-based treatments”. (Page 8, lines 348-350).

Comments 25: Line 385-391: I completely agree with your recommendation for further investigation of MSC treatments of AD. One critical factor that you did not mention was the necessity of adequate funding of the research.

Response 25: We agree with the referee´s comment. These treatments are expensive, and without sufficient funding, their application in companion animals becomes challenging. The next sentence was included in the new version of the manuscript: “However, the need for adequate funding is a critical factor that currently hinders the development of this research, making it complicated and slow”. (Page 9, lines 392-393).

Comments 26: Line 402: Define KCS abbreviation.

Response 26: KCS abbreviation is defined in the first paragraph of the 4.3 Eye diseases section (Page 9, line 402).

Comments 27: Line 427-428: “…with not complete…topical therapy.” Please rewrite for clarity.

Response 27: The sentence has been rewritten for clarity as follows: “Clinical improvement, particularly in tear production, was observed at 3-month follow-up after MSCs treatment, with no further noted improvements. The frequency of administration of topical cyclosporine 2% and artificial tears was progressively reduced by up to half”. (Page 9, lines 427-430).

Comments 28: Line 493: “Other eye diseases…” Please give examples. Are these case studies or case series?

Response 28: Examples of the pathologies seen below (with their abbreviations) have been incorporated in this paragraph. (Page 10, lines 495-496). They are case series, although throughout the bibliography consulted for the current review, many of the investigations in other diseases were carried out in single clinical cases. 

Comments 29: Line 527-536: Same comment as to Line 385-391.

Response 29: Following your suggestion, we have added the following sentence to the end of the paragraph: “As mentioned previously, this will not be possible without adequate funding to support the dissemination of MSCs-based treatments”. (Page 11- lines 539-540).

Comments 30: Line 532: “…microvesicles…” You may want to use extracellular vesicles (EVs) to be consistent with Line 117-118.

Response 30: Corrected. (Page 11- line 535).

Comments 31: Line 537-564: Your discussion in this section is valid, but the reality of the situation is that most of the current application of MSCs is by individual veterinarians on client-owned animals. Case studies and case series studies are considered to be lower-quality research but are valuable research results, none-the-less. Multi-center, double-blind, placebo-controlled clinical trials cannot be done until financing is made available. Autologous use cell therapy does not have a viable financial model. Allogeneic and Xenogeneic use is becoming more common and could have a broader funding appeal to corporate sources of support.

Response 31: We thank referee for this comment, which we completely agree with. We should have indicated this concern in the manuscript. Undoubtedly, a viable financial model is necessary as the first step to conduct multicenter trials and obtain more precise results and protocols. According to the referee's comment, we have included a paragraph highlighting this issue: "Once again, the economic component is a critical factor hindering progress in this field. The lack of an adequate financial model for research and treatment with MSCs make it difficult to conduct multicenter studies and their clinical application, especially in therapies involving autologous cells. Consequently, allogeneic and xenogeneic cell therapies emerge as more reliable alternatives to the autologous route since they require less costs”. (Page 12, lines 571-576).

Comments 32: Line 568: “…inoculation…” Same comment as Line 186.

Response 32: “inoculation” was replaced by “administration” (Page 12, line 587)

Comments on the Quality of English Language

Point 1: There are some areas that have been noted in the comments that are awkward and should be revised. There were a few word choices that may need to be changed.

Response 1: Thank you very much for the comments and suggestions to improve the quality of English language. We have tried to correct, point by point, all these errors in the new version of the manuscript and we have corrected other errors that we found when reviewing the manuscript.

References added in the new version of the manuscript:

  1. Chang, C.; Yan, J.; Yao, Z.; Zhang, C.; Li, X.; Mao, H.Q. Effects of mesenchymal stem cell-derived paracrine signals and their delivery strategies. Adv. Healthc. Mater. 2021, 10, e2001689.

  1. Lin, C.S.; Lin, G.; Lue, T.F. Allogeneic and xenogeneic transplantation of adipose-derived stem cells in immunocompetent recipients without immunosuppressants. Stem Cells Dev. 2012, 21, 2770-2778.

  1. Yu, W.; Wang, J.; Yin, J. Platelet-rich plasma: a promising product for treatment of peripheral nerve regeneration after nerve injury. Int. J. Neurosci. 2011, 121, 176-180.

  1. Abdallah, A.N.; Shamaa, A.A.; El-Tookhy, O.S.; Bahr, M.M. Effect of combined intrathecal/intravenous injection of bone marrow derived stromal cells in platelet-rich plasma on spinal cord injury in companion animals. Open Vet. J. 2021, 11, 270-276.

  1. Giri, J.; Galipeau, J. Mesenchymal stromal cell therapeutic potency is dependent upon viability, route of delivery, and immune match. Blood Adv. 2020, 4, 1987-1997.

  1. Műzes, G.; Sipos, F. Mesenchymal stem cell-derived secretome: A potential therapeutic option for autoimmune and immune-mediated inflammatory diseases. Cells. 2022, 11, 2300.

We thank reviewer 3 for her/his valuable comments on the contents of our manuscript that have contributed to improve its quality. We are keen to incorporate any other suggestion on it.
